# FRELA: FREQUENCY-LAYERED AUDIO WATERMARKING FOR ROBUST CONTENT AUTHENTICATION

## ABSTRACT

In today's world, the published audio content could be easily found at the risk of misuse, including voice cloning or generative model synthesis. This poses ethical and legal threats to individuals, creators, and organizations. Audio watermarking addresses these threats by embedding imperceptible identifiers in the audio. In this paper, we introduce FRELA, a frequency-layered audio watermarking method that distributes the localized watermark across multiple layers of the frequency domain. Each layer is watermarked on the basis of its spectral properties, enabling some to resist audio attacks while others may degrade. This layered redundancy enables partial watermark recovery even when the host signal is distorted. Experimental results demonstrate that FRELA preserves high audio quality while significantly enhancing robustness compared to existing techniques, even under adverse conditions including noise, speed and volume variations, dynamic changes, sampling distortions, temporal modifications, Encodec compression, and most notably, pitch shifting and frequency band filtering. Furthermore, FRELA remains effective in scenarios involving lossy transmission or partial signal degradation, making it suitable for real-world applications in copyright protection and content authentication.

## 1 INTRODUCTION

In the past decade, online audio platforms have grown rapidly. Services such as music streaming applications, podcast hosting platforms, and short-form content sharing have changed how people create, consume, and share audio material. These platforms make it easier for users to share their personal recordings, music, and audio artworks with a wide audience while also providing convenient access to other people's audio content. However, this development introduces risks, particularly in relation to the potential misuse and unauthorized manipulation of shared audio content.

Audio watermarking is a promising mitigation to solve the risk of voice spoofing and audio deepfake. This approach embeds the hidden information, i.e., a "watermark" into an audio signal in a way that is imperceptible to human ears but can be extracted or detected by watermark detectors. It helps to maintain the integrity of the audio sources and protect synthesized speech from being used unethically.

Traditional audio watermarking methods utilize techniques such as LSB (Least Significant Bit) modification (Cvejic & Seppänen, 2004), echo hiding (Gruhl et al., 1996), Fourier transform (Wang & Zhao, 2006), QIM (Quantization Index Modulation) (Chen & Wornell, 2001), and spread spectrum watermarking (Cox et al., 1997). These methods rely on simple mathematical transformations, but they are vulnerable to signal manipulation attacks. In contrast, modern watermarking methods, such as WavMark (Chen et al., 2024), AudioSeal (Roman et al., 2024), and Timbre (Liu et al., 2024), have significantly improved robustness and imperceptibility by leveraging more complex signal processing algorithms that integrate deep neural networks (DNN) or generative adversarial networks (GANs) to embed watermarks in a way that improves transparency while maintaining robustness against various forms of signal distortions.

However, modern watermarking methods still face certain limitations. Particularly, they are vulnerable to synchronization attacks (Wen et al., 2025). Neural network-based watermark detectors are trained in fixed-time alignment assumptions and do not have strong synchronization markers. Therefore, resampling or time-scale modifications such as pitch shift and speed change can easily

desynchronize embedding and the subsequent detection. When the alignment is lost, the embedded watermarks become undetectable. Moreover, these watermarks rely on high-frequency spectral components; that is, many neural network–based watermarking schemes embed bits into narrow, high-frequency bands. This design introduces fragility: pitch shifting can displace the target frequencies, while high-pass or low-pass filtering may remove the watermark entirely.

To address these limitations, motivated by recent watermarking models (Roman et al., 2024) (Abuadbba et al., 2021) and acoustic characteristics in the frequency domain (Constantinescu & Brad, 2023), this work proposes FRELA, a FREquency-LAyered audio watermarking method that embeds localized watermarking in all frequency bands of a speech. FRELA includes two main processes: frequency layering and watermarking. First, the audio waveform is split into multiple distinct frequency ranges. Second, a watermark generator that predicts a corresponding watermark signal from the input frequency bands enables deeper embedding across the entire spectrum, overcoming the limitations of existing models that depend on restricted frequency bands. The detector will detect the watermarks from all layers and analyze them to derive the overall detected result from the watermarked audio. This design increases the precision and robustness of the detection process by distributing the watermark across multiple frequency bands, which makes it more resilient against targeted attacks. Empirically, this is supported by the high detection accuracy and robustness observed under filtering, pitch-shifting, and resampling attacks. We evaluate the effectiveness of FRELA in watermarking speech audios, and compare the results with other models. The results demonstrate the effectiveness of FRELA in protecting speech from voice cloning techniques while maintaining audio quality.

The contributions of this paper are listed as follows.

- We propose FRELA, a frequency-layered audio watermarking method that embeds watermarks across multiple frequency ranges. This design addresses the limitations of prior approaches, which remain susceptible to frequency-targeted attacks such as filtering, pitch shifting, and resampling.

- We demonstrate that FRELA achieves high imperceptibility, with a Scale-Invariant Signal-to-Noise Ratio (SI-SNR) of 24.52, a Perceptual Evaluation of Speech Quality (PESQ) score of 4.08, and a Cosine Similarity of 0.99, while also exhibiting strong robustness across more than 10 common no-box perturbations, achieving a perfect detection rate of 1.00. Compared to state-of-the-art watermarking methods, FRELA consistently outperforms existing approaches, particularly under frequency-domain attacks.

## 2 RELATED WORK

**Imperceptible audio watermarking**. The development of generative models leads to the incorporation of deep learning models in the encoder and decoder architectures of modern watermark methods (Chen et al., 2024)(Roman et al., 2024)(Liu et al., 2024)(Roman et al., 2025), outperforming the traditional way of using domain-specific characteristics (Kalantari et al., 2009)(Lie & Chang, 2006). For example, AudioSeal (Roman et al., 2024) embeds a learned watermark into specific speech time segments, and its detector estimates watermark presence with time-step resolution. This method ensures high robustness against audio distortions and allows multi-bit watermarking for attribution, which we will use in our work. Timbre Watermarking (Liu et al., 2024) operates by transforming speech into frequency domain using Short-Time Fourier Transform (STFT) and embedding a binary watermark across selected frequency bins. This method is repeated along the time axis and trained with a distortion layer to increase the ability of the extractor to recover the watermark after cloning or degradation. However, the partial embedding can leave the audio vulnerable to targeted frequency attacks such as low-pass filtering (Wen et al., 2025). Moreover, the embedding remains static across time and frequency dimensions, without adaptive modulation to the content or frequency bands. It also lacks temporal or spectral localization of the watermark, making it easier for attackers to exploit.

**Invisible vulnerable watermarking model**. Another type of watermarking method (Cheddad et al., 2010), commonly used in images, employs a fragile design in which even minor modifications to the content can invalidate the watermark. Although vulnerable watermarking offers a promising approach to mitigating the risks of voice spoofing and deepfake audio, ensuring sufficient security

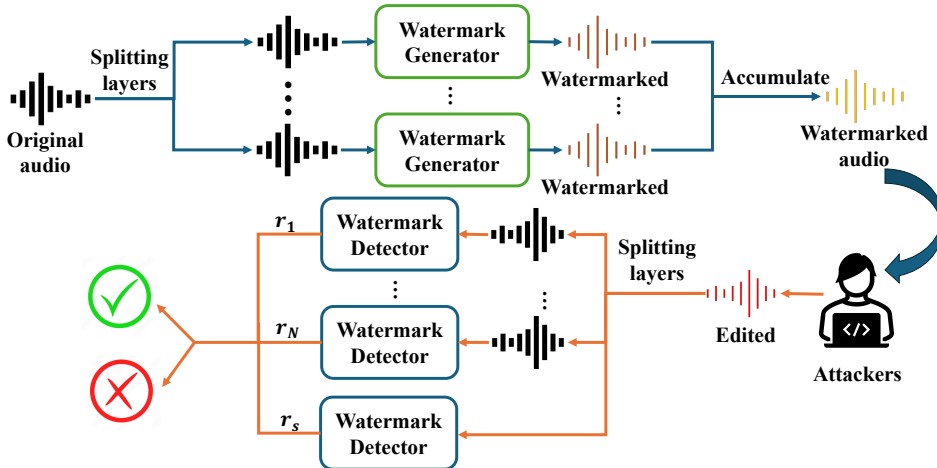

Figure 1: System design of FRELA. The framework consists of two processes: embedding and detection. Input audio is decomposed into multiple frequency bands, where each band is processed by a localized watermark generator. The bandwise watermarked signals are recombined to produce the final watermarked audio. For detection, the audio is split into the same bands, and each band as well as the full audio is evaluated by a detector. A max-pooling decision rule aggregates scores to ensure robustness against frequency-targeted attacks such as filtering, pitch shifting, and resampling.

remains a key challenge. Most existing schemes embed watermarks at fixed positions, making them fragile and easy for attackers to remove. Introducing multiple random embedding positions can help conceal the watermark more effectively. For example, DeepiSign (Abuadbba et al., 2021) applies the concept of invisible vulnerable watermarking within the Convolutional Neural Network (CNN) model itself, embedding a watermark in each hidden layer. This design is fragile, as even small changes to the model can lead to mismatches and trigger tamper detection. In this work, we adapt this idea to audio watermarking, and benchmark our approach against state-of-the-art audio watermarking schemes, including AudioSeal (Roman et al., 2024), WavMark (Chen et al., 2024), and Timbre (Liu et al., 2024).

## 3 METHODS

### 3.1 OVERVIEW

Figure 1 shows the watermarking pipeline. The pipeline consists of two processes: embedding and detection. In the embedding process, we divide the signal into multiple frequency bands. For each band, the magnitude spectrum is isolated and a watermark is embedded. The bands are then recombined and the signal is reconstructed using the Inverse Short-Time Fourier Transform (ISTFT). During detection, the same bandwise process is applied, and the detector is used to recover the embedded watermark.

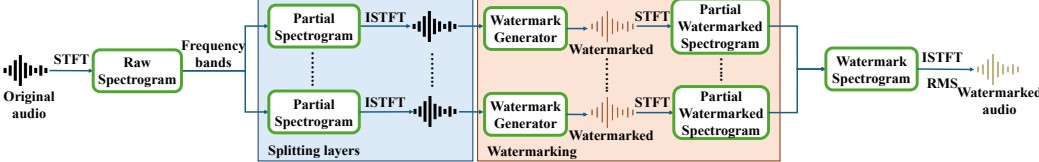

Figure 2: Frequency band decomposition in FRELA. The input spectrogram is divided into multiple contiguous sub-bands, each isolated through band-limited STFT/ISTFT to produce time-domain signals for localized watermark embedding.

Figure 2 shows the overall watermarking pipeline of FRELA. The input audio is first transformed into the frequency domain using STFT and partitioned into multiple sub-bands. Each sub-band is then

converted back to the time domain via ISTFT, where the localized watermark generator produces a band-specific watermark that is embedded into the corresponding signal. The final watermarked audio is reconstructed by combining all band-embedded signals.

## 3.2 Preprocessing and Short-Time Fourier Transform (STFT)

To ensure consistency and compatibility in the data set, a standardized audio preparation is applied to all input files before the watermarking framework. The watermarking model used in this work is trained on single-channel (mono) input. Therefore, signal inputs are converted to mono by computing the of all channels at each time step. Then, all waveforms are resampled to the sampling rate (16 kHz). The sample rate is chosen based on designed model and to balance temporal resolution and computational efficiency

We transform the processed audio signal $x(t)$ into the frequency domain $S(f, t)$ with STFT, which calculates the Fourier transform in short form. At each time step $t$, a segment of $N$ samples is extracted, multiplied by a Hann window $w[n]$. This window tapers the signal at the frame boundaries and reduces spectral leakage. The transformation is represented as follows:

$$w[n] = 0.5 \left( 1 - \cos \left( \frac{2\pi n}{N-1} \right) \right), n = 0, 1, ..., N-1, \tag{1}$$

where $N$ is the frame length and $n$ is the sample index within the frame. The STFT of $x(t)$ is then computed as:

$$S(f, t) = \sum_{n=0}^{N-1} x(t+n) \cdot w[n] \cdot e^{-j\frac{2\pi f n}{N}}, \tag{2}$$

where $x(t+n)$ is the time-domain signal at index $t+n$, $w[n]$ is the Hann window function, $f$ is the discrete frequency index, and $S(f, t)$ is the complex-valued STFT coefficient that represents the amplitude and phase of frequency $f$ at frame $t$.

## 3.3 Divide into Frequency Layers

The STFT is divided into multiple predefined $B$ space sub-bands. Each sub-band $\mathcal{B}_i$ corresponds to a contiguous range of frequency bins, defined as:

$$\mathcal{B}_i = \left( i \cdot \frac{N_{\text{bins}}}{B}, (i+1) \cdot \frac{N_{\text{bins}}}{B} \right), \tag{3}$$

where $N_{bins}$ is the number of frequency bins in the STFT process ($N_{bins} = FFT/2 + 1$). For each band, a band-limited spectral region is isolated by excluding all other frequencies and only preserving the content within the band $\mathcal{B}_i$, as shown below:

$$S_b(f, t) = \begin{cases} S(f, t) & \text{if } f \in \mathcal{B}_i \\ 0 & \text{otherwise.} \end{cases} \tag{4}$$

where $S(f, t)$ is the STFT of the original audio signal, $S_b(f, t)$ is the band-limited STFT for band $\mathcal{B}_i$, and $t$ indexes the time frame. By setting all STFT coefficients outside $\mathcal{B}_i$ to zero, we effectively isolate the spectral content of the band while discarding contributions from other frequencies. Then, each band-limited spectrogram is transformed back into the time domain $x_b(t)$ with the ISTFT, which can be written as:

$$x_b(t) = \text{ISTFT}(S_b(f, t)). \tag{5}$$

In the end, we obtain a set of band-limited time-domain signal $x_b(t)$ to process independently and recombine after watermarking.

## 3.4 Watermarking Model

We use a localized watermarking model based on AudioSeal (Roman et al., 2024). Specifically, we embed a generated watermark in each frequency band. With the input signal $x_b(t)$, a watermark $\omega_b(t)$ created by a generator $\mathcal{G}$ can be expressed as follows:

$$\omega_b(t) = \mathcal{G}(x_b(t)). \tag{6}$$

---

**Algorithm 1** Watermark Detection in Transformed Frequency Bands

---

**Require:** Audio bands $\{b_i\}_{i=1}^{B}$, Full watermarked audio $\tilde{x}$, Detector $\mathcal{D}$, Sample rate $sr$
**Ensure:** Maximum detection score $S$
1: $S \leftarrow 0$ {Initialize detection score}
2: **for** $i = 1$ to $B$ **do**
3:     $b_i' \leftarrow$ Adjust length to match transformation input
4:     $s_i \leftarrow \mathcal{D}(b_i', sr)$ {Run watermark detector on transformed band}
5:     $S \leftarrow \max(S, s_i)$
6: **end for**
7: $\tilde{x}' \leftarrow$ Reshape full watermarked audio to expected input shape
8: $s_{\text{audio}} \leftarrow \mathcal{D}(\tilde{x}', sr)$
9: $S \leftarrow \max(S, s_{\text{audio}})$
10: **return** $S$

---

The watermark $\omega_b(t)$ is added to the original sub-band waveform $x_b(t)$ to obtain the watermarked version $x_b^\omega(t)$, written as follows:

$$x_b^w(t) = x_b(t) + \omega_b(t). \tag{7}$$

We reconstruct the complete audio by applying the same layering process in reverse. STFT transforms each band-limited signal back into the frequency domain $S_b^\omega(f, t)$, as shown below.

$$S_b^\omega(f, t) = \text{STFT}(x_b^\omega(t)). \tag{8}$$

The full watermarked frequency representation $S_{\text{final}}(f, t)$ is reconstructed by adding up all watermarked sub-bands as follows:

$$S_{\text{final}}(f, t) = \sum_b S_b^\omega(f, t). \tag{9}$$

Finally, the inverse STFT recovered the full-band watermarked signal in the time domain waveform $x^\omega(t)$, expressed as follows:

$$x^\omega(t) = \text{ISTFT}(S_{\text{final}}(f, t)). \tag{10}$$

To preserve the scale of loudness and amplitude, we normalize the final watermarked signal $x^\omega(t)$ to match the mean root square (RMS) energy of the original input signal $x(t)$.

$$\text{RMS}(x) = \sqrt{\frac{1}{N} \sum_{i=1}^{N} x(t)^2}, \tag{11}$$

$$x^\omega(t) = x(t) \cdot \frac{\text{RMS}(x(t))}{\text{RMS}(x^\omega(t)) + \varepsilon}, \tag{12}$$

where $\varepsilon$ is a small constant to prevent being divided by zero. The final normalization makes sure the energy of the watermarked output matches the input signal. This preserves the perceptual ability of the watermark and the robustness in the playback environment.

### 3.5 Watermark Detection

To retrieve and validate the watermark, we use the detector $\mathcal{D}$ and the number of frequency bins $B$ in which the watermark is embedded. The detection mirrors the embedding strategy, but instead of adding a watermark to the signal, we recover it from each frequency component of the signal.

As described in Algorithm 1, the watermarked speech signal is first decomposed into $B$ equal frequency bands. Each band $b_i$ is reshaped to match the expected input format of the detector. The detector $\mathcal{D}$ computes a detection score $s_i \in [0, 1]$ for each transformed band. These scores represent the confidence of watermark presence within each band. To ensure robustness, we track the maximum detection score across all bands.

In addition to the band-limited signals, we also process the full-band watermarked audio $\tilde{x}$ to compute a global detection rate $s_{\text{audio}}$. Finally, the overall detection score $S$ is obtained by taking the

maximum between the band-limited and full-band scores. This ensures that even if certain frequency bands are degraded or attacked, the watermark can still be reliably detected from either unaffected bands or the global signal. $S$ can be written as below:

$$S = \max\left(\{\mathcal{D}(\mathcal{T}(b_i), sr)\}_{i=1}^N \cup \{\mathcal{D}(\tilde{x}, sr)\}\right). \tag{13}$$

If $S \geq \tau$, where $\tau \in [0, 1]$ is a detection threshold, then the watermark is considered as "present".

### 3.6 ATTACK DETECTION

In this work, we propose an attack detection mechanism. The hypothesis is that if a watermark embedded in an audio signal becomes undetectable, then the audio has likely been modified. This enables us to use the degradation or absence of a detectable watermark as an indicator of unauthorized or unintended modifications.

Rather than carrying payload information, the watermark functions as a signal integrity marker. Due to the frequency-layered watermark, instead of relying on a single global watermark, this method allows for per-band detection, meaning each sub-band acts as an independent integrity marker. During evaluation, if a subset of bands retain their watermark while others do not, the inconsistency signals that some portion of the signal has likely been tampered with.

At detection time, the watermarked signal $\tilde{x}$ is analyzed to extract detection scores for each band $s_i \in [0, 1]$. Each $s_i$ is compared to a threshold $\tau$ (default = 0.5) to determine if the watermark in band $i$ is still present:

$$\text{Detected}_i = \begin{cases} 1 & \text{if } s_i \geq \tau \\ 0 & \text{otherwise} \end{cases} \tag{14}$$

To determine whether the audio has been modified by attackers, we check for inconsistency across $B$ bands. If not all watermarks are detected, the audio is likely tampered, as shown below:

$$\text{Tampered}(x') = \begin{cases} 1, & \text{if } \sum_{i=1}^B \text{Detected}_i < B \\ 0, & \text{if } \sum_{i=1}^B \text{Detected}_i = B \end{cases} \tag{15}$$

## 4 EVALUATION

### 4.1 AUDIO QUALITY

Table 1 presents the audio quality results on the LJSpeech dataset (Ito, 2017), which contains 13,100 short audio samples. We evaluate FRELA and existing watermarking methods in terms of watermarked audio quality using several metrics: Scale-Invariant Signal-to-Noise Ratio (SI-SNR), Perceptual Evaluation of Speech Quality (PESQ), cosine similarity between raw waveforms and audio embeddings, and Short-Time Objective Intelligibility (STOI). These metrics collectively measure the imperceptibility of the watermark and the overall audio quality, with higher values indicating better performance. The SI-SNR is defined as:

$$\text{SI-SNR} = 10 \cdot \log_{10}\left(\frac{\|s_{\text{target}}\|^2}{\|e_{\text{noise}}\|^2}\right), \quad \text{where} \quad s_{\text{target}} = \frac{\langle \hat{s}, \; s \rangle}{\|s\|^2} \cdot s, e_{\text{noise}} = \hat{s} - s_{\text{target}}.$$

From Table 1, we observe that all methods achieve near-perfect intelligibility and speech consistency (STOI = 0.99 and cosine similarity = 0.99), indicating no significant degradation of the watermarked speech. Among them, FRELA delivers competitive performance across all key metrics. WavMark attains the highest SI-SNR (37.20), reflecting the clearest audio quality, while FRELA achieves a reasonable SI-SNR of 24.83 together with strong perceptual quality (PESQ = 4.04). Notably, the difference in SI-SNR is largely imperceptible to human listeners. Our method thus offers a balanced trade-off between perceptual quality and watermark robustness, as further illustrated in Table 2. AudioSeal provides the best PESQ score (4.08), highlighting its perceptual strength, while Timbre achieves results comparable to AudioSeal in terms of PESQ and STOI.

Table 1: Audio quality results.

| Metrics | SI-SNR↑ | STOI↑ | PESQ↑ | Cosine Similarity↑ |
|---------|---------|-------|-------|--------------------|
| AudioSeal | 25.96 | 0.99 | 4.08 | 0.99 |
| WavMark | 37.20 | 0.99 | 3.99 | 0.99 |
| Timbre | 29.78 | 0.99 | 3.99 | 0.99 |
| **FRELA** | **24.83** | **0.99** | **4.04** | **0.99** |

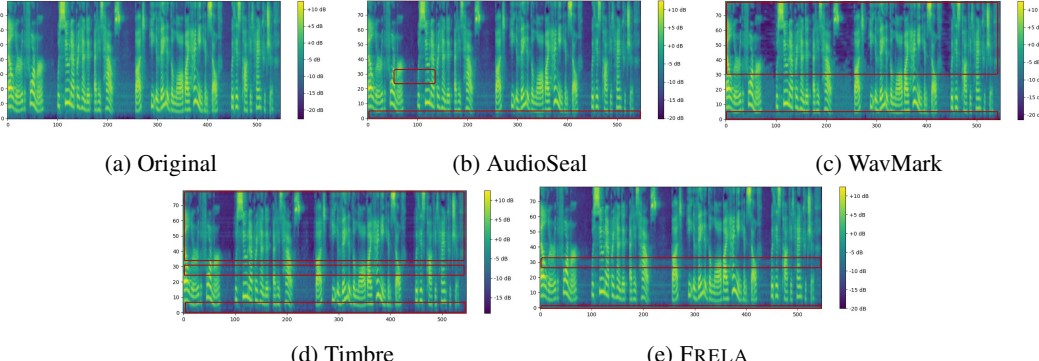

(a) Original  (b) AudioSeal  (c) WavMark

(d) Timbre  (e) FRELA

Figure 3: Watermark audio samples' mel-spectrograms. The red boxes highlight the difference between the watermarked mel spectrogram and the original one, which indicate the spectral changes caused by the embedding process. Clearer differences between the original and watermarked spectrograms correspond to more perceptually noticeable changes for human listeners.

Figure 3 presents mel-spectrogram examples comparing original audio with outputs from different watermarking methods. WavMark and Timbre exhibit the most noticeable distortions, particularly in the high-frequency bins. Timbre further introduces artifacts across multiple frequency ranges due to its use of binary watermark embedding in selected frequencies. In contrast, AudioSeal and FRELA show only minor deviations from the original spectrogram, with slight changes in the low-frequency region. Since FRELA builds on AudioSeal's generator model, it produces similar patterns but with more variation in the mid-frequency range. This difference arises from our strategy of embedding across multiple frequency bands rather than restricting to a narrow set.

## 4.2 ROBUSTNESS AGAINST AUDIO ATTACKS

We evaluate robustness using 200 non-watermarked audio samples from the LJSpeech dataset (Ito, 2017). Following the same attack and evaluation procedure as AudioSeal (Roman et al., 2024), each watermarking method is applied to generate 200 watermarked audio samples, which are then subjected to a variety of transformations. For detection, we vary the decision threshold $\tau$: if the detection score exceeds $\tau$, the watermark is considered present. Table 2 reports the maximum detection accuracy achieved over the range of thresholds. In addition, we measure the True Positive Rate (TPR), which reflects correct detections of watermarked audio, the False Positive Rate (FPR), which reflects incorrect detections on original audio, and the overall accuracy rate (TPR/FPR). AUC summarizes the overall discriminative power of the detector across varying thresholds by showing the trade-off between TPR and FPR.

Table 2 reports the performance of FRELA, AudioSeal, and WavMark under a wide range of no-box perturbations. Across 21 transformations, including clean audio (none), additive noise, filtering (low-pass, high-pass, band-pass), speed and pitch edits (0.8x slower, 1.2x faster, pitch shift), volume and dynamics (boost, ducking, echo, masking, pink noise, white noise, smoothing), resampling, downsampling, temporal edits (cropping, concatenation), and Encodec compression (with and without smoothing), FRELA achieves a perfect detection rate in all categories.

AudioSeal performs reliably under simple edits (e.g., additive noise, echo, volume boost) but degrades sharply under spectral and pitch modifications, with accuracy dropping to 0.52 on pitch

Table 2: **Detection results**. Performance of FRELA in comparison with watermarking methods (Chen et al., 2024)(Roman et al., 2024)(Liu et al., 2024) under various audio attacks.

| Attacks | AudioSeal Acc | TPR/FPR | AUC | WavMark Acc | TPR/FPR | AUC | Timbre Acc | TPR/FPR | AUC | FRELA Acc | TPR/FPR | AUC |
|---|---|---|---|---|---|---|---|---|---|---|---|---|
| None | 1.00 | 1.00/0.00 | 1.00 | 1.00 | 1.00/0.00 | 1.00 | 1.00 | 1.00/0.00 | 1.00 | 1.00 | 1.00/0.00 | 1.00 |
| Noise | 1.00 | 0.99/0.00 | 1.00 | 0.51 | 0.01/0.00 | 0.51 | 1.00 | 1.00/0.00 | 1.00 | **1.00** | **1.00/0.00** | **1.00** |
| Low-pass | 1.00 | 1.00/0.00 | 1.00 | 1.00 | 1.00/0.00 | 1.00 | 1.00 | 1.00/0.00 | 1.00 | 1.00 | 1.00/0.00 | 1.00 |
| High-pass | 0.54 | 0.28/0.20 | 0.52 | 0.73 | 0.47/0.00 | 0.73 | 0.92 | 0.97/0.14 | 0.95 | **1.00** | **1.00/0.00** | **1.00** |
| Band-pass | 0.53 | 0.54/0.47 | 0.51 | 0.98 | 0.97/0.00 | 0.98 | 1.00 | 1.00/0.00 | 1.00 | **1.00** | **1.00/0.00** | **1.00** |
| Speed (0.8x) | 0.96 | 0.94/0.02 | 0.99 | 1.00 | 1.00/0.00 | 1.00 | 1.00 | 1.00/0.00 | 1.00 | 1.00 | 1.00/0.00 | 1.00 |
| Speed (1.2x) | 1.00 | 1.00/0.00 | 1.00 | 0.99 | 0.99/0.00 | 0.99 | 1.00 | 1.00/0.00 | 1.00 | 1.00 | 1.00/0.00 | 1.00 |
| Pitch shift | 0.52 | 0.12/0.07 | 0.52 | 0.50 | 1.00/1.00 | 0.50 | 0.60 | 0.99/0.81 | 0.50 | **1.00** | **1.00/0.00** | **1.00** |
| Boost | 1.00 | 1.00/0.00 | 1.00 | 1.00 | 1.00/0.00 | 1.00 | 1.00 | 1.00/0.00 | 1.00 | 1.00 | 1.00/0.00 | 1.00 |
| Duck | 1.00 | 1.00/0.00 | 1.00 | 1.00 | 1.00/0.00 | 1.00 | 1.00 | 1.00/0.00 | 1.00 | 1.00 | 1.00/0.00 | 1.00 |
| Echo | 1.00 | 1.00/0.00 | 1.00 | 1.00 | 1.00/0.00 | 1.00 | 0.99 | 1.00/0.01 | 0.99 | 1.00 | 1.00/0.00 | 1.00 |
| Mask | 1.00 | 1.00/0.00 | 1.00 | 0.65 | 0.30/0.00 | 0.65 | 0.99 | 0.99/0.00 | 0.99 | **1.00** | **1.00/0.00** | **1.00** |
| Pink | 1.00 | 1.00/0.00 | 1.00 | 0.50 | 0.01/0.00 | 0.50 | 0.98 | 0.99/0.04 | 0.99 | 1.00 | 1.00/0.00 | 1.00 |
| White | 0.99 | 0.97/0.00 | 0.99 | 0.50 | 1.00/1.00 | 0.50 | 0.94 | 0.95/0.16 | 0.95 | 1.00 | 1.00/0.00 | 1.00 |
| Smooth | 1.00 | 1.00/0.00 | 1.00 | 0.50 | 0.50/0.50 | 0.50 | 1.00 | 1.00/0.00 | 1.00 | 1.00 | 1.00/0.00 | 1.00 |
| Resampled | 0.50 | 0.00/0.00 | 0.44 | 0.50 | 0.00/0.00 | 0.50 | 0.87 | 0.98/0.25 | 0.88 | **1.00** | **1.00/0.00** | **1.00** |
| Downsampled | 0.51 | 0.21/0.19 | 0.47 | 0.50 | 0.00/0.00 | 0.50 | 0.84 | 0.93/0.25 | 0.87 | **1.00** | **1.00/0.00** | **1.00** |
| Cropped | 1.00 | 1.00/0.00 | 1.00 | 0.99 | 0.98/0.00 | 0.99 | 1.00 | 1.00/0.00 | 1.00 | 1.00 | 1.00/0.00 | 1.00 |
| Concat | 1.00 | 1.00/0.00 | 1.00 | 1.00 | 1.00/0.00 | 1.00 | 1.00 | 1.00/0.00 | 1.00 | 1.00 | 1.00/0.00 | 1.00 |
| Encodec (w/o smooth) | 0.52 | 0.36/0.32 | 0.51 | 0.50 | 0.50/0.50 | 0.50 | 0.50 | 0.97/0.96 | 0.37 | **1.00** | **1.00/0.00** | **1.00** |
| Encodec (with smooth) | 0.90 | 0.85/0.06 | 0.95 | 0.50 | 0.50/0.50 | 0.50 | 0.61 | 0.85/0.64 | 0.63 | **1.00** | **1.00/0.00** | **1.00** |
| Average | 0.86 | 0.77/0.06 | 0.85 | 0.75 | 0.68/0.17 | 0.75 | 0.90 | 0.92/0.15 | 0.91 | **1.00** | **1.00/0.00** | **1.00** |

shifting and 0.53 on band-pass filtering. Its performance also suffers under Encodec compression without smoothing (Acc = 0.52). By contrast, FRELA embeds watermarks more deeply across the spectrum, maintaining flawless robustness even against spectral and pitch-based attacks.

Timbre ranks as a strong second-best performer (Acc = 0.90, TPR = 0.92, AUC = 0.91). In contrast, WavMark suffers severe degradation under most perturbations. Despite achieving the highest audio quality in Table 1, its fragile watermark embedding renders it unsuitable for adversarial or uncontrolled environments.

Overall, FRELA consistently outperforms all baselines across all three evaluation metrics. It achieves a 100% detection rate with no false positives or false negatives. Its high AUC further reflects robustness to threshold variation and confidence estimation, even under heavily degraded conditions. Importantly, this robustness is achieved without compromising perceptual quality: FRELA maintains high PESQ and SI-SNR while ensuring reliable detection. This advantage arises from its design choice of embedding watermarks across multiple frequency bands rather than confining them to specific bins, ensuring that the watermark remains intact even when attacks target only parts of the spectrum.

## 4.3 ATTACK DETECTION PERFORMANCE

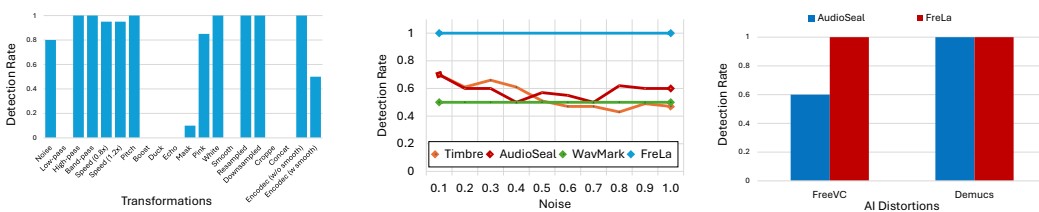

Figure 4: Attack detection rate.          Figure 5: Partial signal attacks          Figure 6: AI distortions

FRELA also enables detection of whether an audio signal has been modified. Detection scores close to 1.0 indicate that FRELA correctly identifies the presence of modifications. As shown in Figure 4, FRELA effectively detects a wide range of common perturbations, including band-pass and high-pass filtering, pitch and speed changes (0.8x, 1.2x), white noise, resampling, downsampling, Encodec compression, and concatenation. Additive noise attacks (standard noise, pink noise) are also largely detected, with scores in the 0.8–0.9 range, demonstrating the system's resilience

in noisy environments. However, temporal-structure-based attacks such as cropping, echoing, and partial muting are able to bypass detection. This limitation arises because such attacks primarily disrupt local temporal continuity rather than altering the spectral content across frequency bands, making them less visible to frequency-oriented watermark detectors. Addressing these vulnerabilities may require integrating complementary time-domain analysis or hybrid watermarking strategies to strengthen robustness against temporally localized manipulations.

As shown in Figure 5, FRELA demonstrates strong robustness against partial degradation. The detection rate remains stable at 1.00 across all tested noise levels, indicating that the watermark is unaffected even under increasingly severe perturbations. In contrast, baseline methods such as Timbre, AudioSeal, and WavMark exhibit a clear decline in detection performance as the noise level increases, highlighting their vulnerability to additive distortions.

We further evaluate the robustness of FRELA under AI-induced distortions: Demucs (Rouard et al., 2023)(Défossez, 2021) and FreeVC (li et al., 2022). Demucs is a music source separation model that reconstructs audio into stems and introduces reconstruction artifacts, while FreeVC is a voice conversion model that resynthesizes speech and alters speakers' voices. Figure 6 shows that FRELA achieves a detection rate of 100%, outperforming AudioSeal.

## 5 DISCUSSION

**Limited evaluated dataset.** Although we apply our watermarking method to 13,100 audio samples. Attacks are tested only on a subset of 200 samples. This restricted evaluation may not fully capture the performance and robustness of different methods. In addition, the data set consists primarily of short audio clips ranging from 1 to 10 seconds. In real-world scenarios, audio content often includes much longer recordings. In future work, we plan to expand our evaluation to larger and more diverse datasets, including long-form audio, to better assess the effectiveness of our approach.

**Limited attack detection results.** Current watermarking methods focus mainly on indicating whether a piece of audio has been reused in other content, but cannot reliably detect whether the audio has been modified. Our method introduces a lossy watermarking scheme that preserves the watermark only in regions of the audio that have not been heavily altered, enabling users to identify when their audio has been tampered with. However, this detection capability is limited to certain types of attack, as shown in Figure 4. In future work, we will expand our investigation to improve robustness and broaden the range of detectable modifications.

**Existing Watermark Methods.** To evaluate the efficiency of our scheme, we perform attacks on several existing watermarking methods. As shown in Table 2, most existing watermarking approaches do not resist filtering, pitch shifting, and resampling attacks. This weakness stems from their reliance on high-frequency bands and strict time alignment assumptions. The same trend is illustrated in Figure 3, except for AudioSeal and FRELA, other methods indicate a significant difference in the high-frequency regions of their mel spectrograms.

## 6 CONCLUSION

In this paper, we propose FRELA, a frequency-layered audio watermarking method. Unlike existing approaches, FRELA embeds localized watermarks in multiple frequency bands of speech. This design enables the watermark to persist under partial signal degradation or lossy transmission. Moreover, after modifications, the watermark can still be detected and used to identify whether the audio has been altered. Through experiments, FRELA demonstrates strong robustness against a wide range of modifications and provides the ability to detect many types of attack. Beyond demonstrating its practical robustness, FRELA highlights the value of frequency-layered embedding as a general strategy for resilient watermarking. By distributing watermark information across multiple bands, FRELA avoids the fragility of single-band methods and ensures graceful degradation under adversarial conditions. While our results show excellent performance, certain temporal-structure-based attacks (e.g., cropping, echo, muting) remain challenging, pointing to future work in hybrid watermarking that combines spectral and temporal localization. In addition, applying FRELA to diverse domains such as music, streaming platforms, and real-time communication opens new opportunities for protecting digital audio integrity at scale.

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
