# OpenReview forum: "FRELA: Frequency-Layered Audio Watermarking for Robust Content Authentication"
_ICLR.cc/2026/Conference — Submitted to ICLR 2026_

### Official Review · Reviewer_xbdX · 2025-10-29

**Soundness:** 3
**Presentation:** 2
**Contribution:** 1
**Rating:** 2
**Confidence:** 4

**Summary:**

This paper proposed a 1-bit watermark called FRELA, which splits the target audio into disjoint but contiguous sub-bands then applies AudioSeal to each sub-band signal. The proposed scheme can also be used as an integrity check to indicate if the signal has been modified or attacked by certain types of attacks.

**Strengths:**

Proposed layered redundancy has potential to be adapted for other watermarks than AudioSeal.
Moreover, this redundancy can be also used in integrity check via observing partial survival of watermark in each sub-band.
The methodology part is easy to follow at high abstraction level, despite some important detail seems to be missing.
Despite applying AudioSeal for multiple times, the quality degradation is still maintained in an acceptable range (1.13db SISNR, 0.04 PESQ).

**Weaknesses:**

Since proposed method focuses on detecting the presence of watermark, it is not the same watermarking scheme as WavMark, Timbre or AudioSeal. Those watermarks are designed for hiding specific message (>10 bits) within an audio. In contrast, proposed method is equivalent to 1-bit watermarking (watermarked or not), claiming superiority over those works is not proper since the problem setups are different. A fair comparison should be made to other 1-bit watermarking, or refine the statements in the introduction.

Another concern is the lack of detail, which impacts reproducibility and forbids future investigation from other researchers. For example, there's no information about parameter B, N, nor the STFT hopsize. The information on the detector is also lacking. Although readers can infer that it's probably a classifer network, but that's all and nothing more.

The detail about evaluation is also lacking. While in L362 it says "Following the same attack and evaluation procedure as AudioSeal". However, the detection score in original AudioSeal included computing the hamming distance between the recovered message and the original message, which is clearly not an 1-bit scenario. Moreover, some types of attack are not exist in AudioSeal (e.g. "Mask", "Pitch Shift", "ReSample", "Downsample", "Crop", "Concat"). Some of the attack parameters are even different than AudioSeal (e.g. the speed up is 1.25x in AudioSeal, but 0.8/1.2x in the evaluation of proposed work). In a later section(sect 4.3, L438), it suddenly mentioned about "partial degradation" without explaining what it is.

In the end, the lack of information makes the manuscript and evaluation result unconvincing due to missing necessary information. This also prevents any reader from verifying or gaining reusable insights.

**Questions:**

As I mentioned in the "weakness" part, would like to know the following details:
- The information on the detector
- Hyperparameter setups (N, B, hopsize, etc.)
- The parameters and algorithmic steps of each attacks in the evalutation.

---

### Official Review · Reviewer_5vKD · 2025-10-30

**Soundness:** 1
**Presentation:** 2
**Contribution:** 2
**Rating:** 2
**Confidence:** 3

**Summary:**

The paper proposes FRELA. The approach first applies STFT to the speech signal and divides the resulting spectrum into multiple adjacent sub-bands. Within each sub-band, a local watermark generator independently embeds a watermark. During detection, scores are computed for each sub-band as well as for the entire audio clip, and the final decision is obtained through max-pooling aggregation. The authors claim that this strategy maintains robustness when the signal undergoes partial spectral distortion (such as band-pass, high-pass, or low-pass filtering) or desynchronization attacks (such as time-stretching, pitch-shifting, or resampling), while preserving high perceptual quality. Experiments conducted on LJSpeech compare FRELA with AudioSeal, WavMark, and Timbre. Table 1 reports SI-SNR, PESQ, STOI, and cosine similarity under no-attack conditions, while Table 2 presents detection results under more than 20 perturbations, claiming that accuracy, TPR, and AUC all reach 1.00.

**Strengths:**

The paper accurately identifies the vulnerability of modern audio watermarking systems to frequency-domain attacks, which represents a valuable research direction.

The core idea of distributing watermark information across the entire spectrum is conceptually sound and provides a logical path toward improving robustness.

The writing of the paper is very clear, making the work easy to follow.

**Weaknesses:**

Table 2 shows nearly all results as 1.0; according to the description, the values were obtained by threshold sweeping to find the optimal point, suggesting a possible selective or overly optimistic bias. A global single threshold should be used instead, and ROC/AUC, TPR@FPR=1%/0.1%, EER, and 95% confidence intervals should be reported, along with per-attack ROC curves.


Factors such as the number and division of sub-bands (equal-width vs. perceptually equivalent), max vs. average/weighted pooling, cross-band weight sharing or conditioning, embedding strength λ, and STFT parameters are not systematically presented, making it difficult to identify the truly effective components.


Only one dataset and short speech segments are used; tests lack MP3/AAC/Opus at multiple bitrates, platform transmission chains, temporal structure attacks, and combined perturbations, resulting in weak generalizability.


No code, models, scripts, examples, or random seeds are provided; mean ± variance, confidence intervals, and sample size details are also missing.


The work appears more like a system-level integration, lacking mechanistic analysis to support its contribution.


Despite noticeably lower SI-SNR, the paper dismisses this with a brief claim of imperceptibility, without subjective listening tests such as MOS, MUSHRA, or ABX for validation.

**Questions:**

Did you adjust the threshold separately for each type of attack? Please redo the core evaluation using a single global fixed threshold, and report ROC/AUC, TPR at fixed FPR levels, EER, 95% confidence intervals, and per-attack ROC curves.

Have you considered evaluating FRELA on more diverse audio types or with more common codecs such as MP3 and AAC to strengthen the generalizability of your conclusions?

Could you include results across different datasets, codec variations, platform transmission chains, and temporal or combined perturbations?

Would you consider boldface formatting only for the best values within each column (or removing it entirely), releasing your code, models, evaluation scripts, examples, and random seeds, and clarifying how you prevent data duplication or leakage?

---

### Official Review · Reviewer_Vhqk · 2025-10-31

**Soundness:** 2
**Presentation:** 2
**Contribution:** 1
**Rating:** 2
**Confidence:** 4

**Summary:**

This paper proposes an audio watermarking method named FRELA, which enhances the robustness and imperceptibility of the watermark by embedding it in multiple frequency layers of the audio. Mainly, it extends and improves the existing advanced audio watermarking method AudioSeal, enhancing its robustness.

**Strengths:**

The paper is well-written with clear structure. The experimental design comprehensively covers various audio perturbations, and results demonstrate stable performance on the tested datasets, validating the method's robustness.

**Weaknesses:**

The paper proposes FRELA, a frequency-stratified watermarking approach that distributes watermarks across multiple frequency bands to enhance robustness against frequency-domain attacks. However, this approach essentially extends existing methods (e.g., AudioSeal) without significant theoretical innovation. Frequency stratification and multi-band embedding have been widely adopted in image and watermarking literature; the paper fails to introduce methodologically novel mechanisms or breakthroughs.

More critically, while the authors showcase performance improvements through experimental evaluation, the paper suffers from a glaring absence of rigorous theoretical analysis.

Specifically, there is no substantive theoretical justification for how the multi-frequency-layer embedding mechanism enhances robustness against various attacks or maintains audio quality.

**Questions:**

How does multi-band embedding preserve audio quality? The trade-offs between watermark strength, perceptual quality, and robustness across different bands need theoretical grounding.

Can the authors propose a principled framework for multi-band watermarking, rather than simply adapting existing single-band models?

In the watermark signal normalization step (Equation 12), should the left-hand side be x^{w}(t) rather than x(t)? The current notation appears inconsistent with the normalization objective.

---

### Official Review · Reviewer_GehE · 2025-11-01

**Soundness:** 3
**Presentation:** 3
**Contribution:** 3
**Rating:** 6
**Confidence:** 4

**Summary:**

This paper introduces FRELA, a Frequency-Layered audio watermarking method that embeds localized watermarks across multiple frequency bands of audio signals. Unlike traditional approaches that confine watermarks to specific frequency regions, FRELA distributes watermark information across the entire spectrum, enabling partial recovery even when portions of the signal are distorted. The method first decomposes input audio into multiple frequency bands, embeds band-specific watermarks using a generator model, and then recombines the watermarked bands. This design significantly enhances robustness against frequency-targeted attacks such as filtering, pitch shifting, and resampling while maintaining high audio quality. The authors demonstrate FRELA's superior performance through comprehensive evaluations against state-of-the-art watermarking approaches, showing perfect detection rates across various audio manipulations and distortions.

**Strengths:**

1. Novel approach: The frequency-layered embedding strategy effectively addresses a critical limitation of existing watermarking methods that typically rely on narrow frequency bands. This represents a meaningful architectural contribution to audio watermarking.

2. Exceptional robustness: FRELA demonstrates impressive resilience against a comprehensive range of audio attacks, achieving perfect detection rates (1.00) across all 21 tested transformations, significantly outperforming AudioSeal, WavMark, and Timbre. The method's robustness to frequency-domain attacks (filtering, pitch-shifting) is particularly noteworthy.

3. Audio quality preservation: Despite distributing watermarks across multiple frequency bands, FRELA maintains high audio quality with competitive SI-SNR (24.83), PESQ (4.04), and perfect STOI (0.99) scores, demonstrating a good balance between robustness and imperceptibility.

4. Attack detection capability: The proposed method not only embeds recoverable watermarks but also enables detection of whether audio content has been tampered with, which adds a valuable security layer for content authentication applications.

5. Comprehensive evaluation: The experimental evaluation is thorough, covering multiple aspects (quality, robustness, attack detection) across various audio transformations and comparing against several state-of-the-art baselines. The evaluation methodology is clear and the metrics are appropriate.

**Weaknesses:**

1. Limited dataset evaluation scope: While the method was applied to 13,100 audio samples for quality assessment, attack evaluations were limited to only 200 samples as acknowledged by the authors. This relatively small evaluation set may not fully capture real-world performance variations.

2. Short-duration audio focus: The evaluation primarily uses short audio clips (1-10 seconds), whereas many real-world applications involve much longer recordings. It's unclear how the method scales to longer content or whether performance degrades over extended durations.

3. Incomplete attack detection: As shown in Figure 4, FRELA shows limitations in detecting temporal-structure-based attacks like cropping, echoing, and partial muting. This reduces its effectiveness as a comprehensive content authentication solution.

4. Computational complexity analysis: The paper lacks discussion of computational requirements and efficiency, which is crucial for practical deployment, especially in resource-constrained environments or real-time applications.

5. Limited theoretical foundation: While the empirical results are strong, the paper could benefit from deeper theoretical analysis explaining why the frequency-layered approach provides superior robustness compared to existing methods.

**Questions:**

1. How does the computational complexity of FRELA compare to existing methods like AudioSeal or Timbre? Is the method suitable for real-time applications on devices?

2. The paper mentions limitations in detecting temporal-structure-based attacks (cropping, echo, muting). Have you explored any extensions to the frequency-layered approach that might address these limitations, perhaps by incorporating temporal localization?

3. How does the performance scale with longer audio recordings? Were there any experiments conducted on audio samples longer than 10 seconds, and if so, did you observe any degradation in watermark robustness?

4. How was the number of frequency bands (B) determined? Is there an optimal number of bands for different types of audio content, and how does this parameter affect the trade-off between robustness and audio quality?

5. Could you provide more details about how the method might perform in cascaded attack scenarios where multiple transformations are applied sequentially (e.g., pitch shifting followed by compression)?

---

### Meta-Review · Area_Chair_Nith · 2026-01-04

**Summary:**

This paper proposes FRELA, a 1-bit frequency-layered audio watermarking method that embeds watermarks across multiple sub-bands of an audio signal to improve robustness against frequency-domain and partial-signal attacks while maintaining high audio quality. The method applies STFT to split the audio into contiguous sub-bands, embeds watermarks independently in each band using a generator (based on AudioSeal), and aggregates detection scores via max-pooling. Experiments on LJSpeech show high robustness under various perturbations and minor quality degradation. Strengths include clear writing, a conceptually sound layered redundancy approach, potential for integrity checking, and stable high-level methodology. Weaknesses involve limited novelty (essentially extending AudioSeal to multiple bands), a focus on 1-bit watermarking unlike full-message baselines, insufficient theoretical justification, incomplete methodological and evaluation details (e.g., STFT parameters, detector structure, attack settings), small and short-duration datasets, lack of code or reproducibility information, and evaluation choices that may overestimate performance, limiting both reproducibility and generalizability.

**Reviewer Concerns:**

The authors have not provided a response.

**Reviewer Scores:**

The reviewer will not revise their comments, as the authors have not responded to any of the issues raised.

---

### Decision · Program_Chairs · 2026-01-26

Reject